# eXplainable Artificial Intelligence (XAI) for improving organisational regility

**Niusha Shafiabady**[1]*, **Nick Hadjinicolaou**[2], **Nadeesha Hettikankanamage**[3], **Ehsan MohammadiSavadkoohi**[1], **Robert M. X. Wu**[4], **James Vakilian**[1]

**1** Faculty of Science and Technology, Charles Darwin University, Haymarket, New South Wales, Australia, **2** Adelaide Institute of Higher Education, Adelaide, South Australia, Australia, **3** Design and Creative Technology, Torrens University Australia, Adelaide, South Australia, Australia, **4** Faculty of Engineering and Information Technology, University of Technology Sydney, Broadway, New South Wales, Australia

* niusha.shafiabady@cdu.edu.au

**Data Availability Statement:** Data cannot be shared according to the ethics approval ref EAN 2015 09 (6) from Torrens University HEC. I have attached the ethics approval to the submission for

## Abstract

Since the pandemic started, organisations have been actively seeking ways to improve their organisational agility and resilience (regility) and turn to Artificial Intelligence (AI) to gain a deeper understanding and further enhance their agility and regility. Organisations are turning to AI as a critical enabler to achieve these goals. AI empowers organisations by analysing large data sets quickly and accurately, enabling faster decision-making and building agility and resilience. This strategic use of AI gives businesses a competitive advantage and allows them to adapt to rapidly changing environments. Failure to prioritise agility and responsiveness can result in increased costs, missed opportunities, competition and reputational damage, and ultimately, loss of customers, revenue, profitability, and market share. Prioritising can be achieved by utilising eXplainable Artificial Intelligence (XAI) techniques, illuminating how AI models make decisions and making them transparent, interpretable, and understandable. Based on previous research on using AI to predict organisational agility, this study focuses on integrating XAI techniques, such as Shapley Additive Explanations (SHAP), in organisational agility and resilience. By identifying the importance of different features that affect organisational agility prediction, this study aims to demystify the decision-making processes of the prediction model using XAI. This is essential for the ethical deployment of AI, fostering trust and transparency in these systems. Recognising key features in organisational agility prediction can guide companies in determining which areas to concentrate on in order to improve their agility and resilience.

## 1. Introduction

The significance of organisational agility has seen a tremendous rise since the outbreak of the COVID-19 pandemic [1, 2]. While the pandemic has created opportunities to accelerate the development and use of technology, it has also created a more volatile and uncertain business environment that has forced organisations to become more agile and resilient to survive [3–5]. Organisational agility is "the capability to quickly sense and adapt to external and internal

your reference. The Ethics committee's contact details are: Phone: +61 8 8113 7842 E-mail: ethics@tua.edu.au Quote the reference no: EAN 2015 09 (6).

**Funding:** The author(s) received no specific funding for this work.

**Competing interests:** The authors have declared that no competing interests exist.

changes to deliver relevant results productively and cost-effectively" [6, 7]. Organisational resilience, on the other hand, is defined as "the ability of an organisation to anticipate, prepare for, respond to, and recover from disruptions" [8]. It measures how well an organisation can withstand, rebound from, or even advance in facing challenges.

Therefore, Organisational regility could be referred to as "the ability of an organisation to anticipate, prepare for, respond to, and recover from disruptions" and "the capability to sense and adapt to external and internal changes quickly" [9–11]. In essence, in the same way, there are "defense and offense" strategies in sports. One is recovering from organisational disruptions, whilst the other is looking to take advantage of the opportunities.

Regility is closely linked to the execution of an organisation's agility strategy. Benefits of organisational agility include improved engagement and retention of talent, acceleration of organisational learning, and increased responsiveness when making business decisions [12–15]. Organisational resilience includes having a strong culture of learning and adaptation, a clear sense of purpose, strong leadership, and adequate resources to withstand disruptions [16].

Resilient organisations are willing to learn from their mistakes and adapt to change. They have a clear sense of purpose and influential leaders who provide clear direction by setting positive and motivated employees [17]. Good communication is essential for ensuring that everyone in the organisation knows the risks and challenges and is prepared to respond accordingly. Resilient organisations also have the resources to withstand disruptions, such as financial, human, and technological resources [18–20].

Organisations face the twofold challenge of meeting ever-increasing customer demands and safeguarding their reputations. In this context, using Artificial Intelligence (AI) is becoming increasingly crucial to deal with more significant customer requirements and levels of customer satisfaction required to prevent reputation loss.

AI is employed to personalise client experiences, predict customer requirements, and furnish customer assistance [21, 22]. It can also be deployed to enhance decision-making and make organisations more agile in the face of market contests, legislative modifications, and economic risks. Agile organisations will attract and retain top talent, learn quickly, and respond quickly to changes in the environment [23–25]. This gives them a competitive advantage and allows them to sustain their success when making business decisions [23–25].

AI advancements in the last decade have paved the way for an innovative and more digitalised society. AI plays a pivotal role in various facets of humans in daily life, industry, and business [26–29]. The utilisation of AI for business and industrial applications has been commonplace in previous years [19–23]. Following this evolution, more research is being conducted into AI. Since AI makes systems capable of learning from experience, which can take the future to a higher level of independence, this could be revolutionary. Besides, AI-based methods are becoming more precise, descriptive, and informative and contribute professionals to understanding how decisions and instructions are made [30].

Researchers suggest that IT has a positive impact on organisational agility [31–33]. The accelerated utilisation of AI in organisational domains resulted in the new notion of AI capability, which has been presented to interpret the organisations' resource allocation [34, 35].

[34] evaluates the effect of artificial intelligence capability on organisational agility. They designed a structural model and experimented with it on an industrial sample. Results indicated the great positive impact of AI on organisational agility. All the previous papers assessed organisational agility and the effect of IT and AI on it.

However, the prediction of organisational agility which is a real-world application of AI in this field was still unknown. [12] predicted the organisational agility for data gathered from 44 respondents including private and public organisations in the industry sector. Different

machine learning methods have been employed as predictors and the Random forest method had the best performance with a 97.67% accuracy.

Despite all the efforts in this field, there is an unanswered question when we predict organisational agility. How do we trust AI in predicting a real-world problem with numerous inputs that might have a great impact on organisations' business performance? To answer this question, we need to have a metric to make the organisational agility prediction transparent and explainable. Therefore, extensive investigations should be conducted to discover the effects of different features involved in the problem and to authenticate trustworthiness and transparency because of the intricacy of AI methods. In other words, lack of transparency made the achievements in AI field untrustworthy and unreliable [36–38]. The system behaves like a black-box with no explanation or justification for its predictions [37].

Explainability is the ability to comprehend how and why an AI system comes to decisions. This paper will illustrate the importance of explainability for AI. The goal of eXplainable artificial intelligence (XAI) is to interpret an agent's decisions and help the agent to have a notion of a black-box system, transforming the black-box system into an interpretable and trustworthy white-box system [38, 39].

Furthermore, AI systems are utilised to make critical decisions that commonly have profound consequences for society and industry. Explainability assures that the decision-making procedure is unbiased and fair, and customers have the chance to challenge the decisions they assume to be wrong or discriminated against [39, 40].

As mentioned earlier, explainability is crucial in building confidence in an AI system. In domains such as healthcare, finance, and business, where judgments made by AI systems have a notable effect on individual lives, trust is crucial for using AI [41–43]. If people do not comprehend how an AI system functions or why it is making particular decisions, they are less likely to trust it. By explaining the judgments made by AI systems, we can develop trust and improve the approval for these systems.

Furthermore, explainability is substantial to improve the performance of AI systems. The areas where the system may be making errors or biases will be identified by understanding the reason for making decisions [44, 45]. This lets us clarify the procedure and enhance its accuracy and reliability. Without an explainability concept, it is challenging to recognise and correct the errors in AI systems, which can lead to disastrous aftermaths. Recently, the research into XAI, where the purpose is to perceive the internal workings of a black box, find out which input features mostly influence the outcomes or outputs in a system, and ultimately make the system explainable and interpretable [46]. There are a variety of well-established XAI methods and techniques, such as Local Interpretable Model-agnostic Explanations (LIME) [47], GRADient Class Activation Mapping (GRAD-CAM) [48], Shapley Additive exPlanation (SHAP) [49], Deep Learning Important FeaTures (DeepLIFT) [50] have been employed for prediction tasks across various domains.

Among the spectrum of AI interpretability, the SHAP is the widely adopted and most all-encompassing technique for interpreting feature importance and interaction among various available methods [51]. SHAP is a model-agnostic and has been presented to apply to any machine learning algorithm as the models generate interpretations after training [50–52]. It can be applied to any machine learning model, disregarding its architecture or tutoring algorithm [50–53]. This method enhances the interpretability of pre-existing machine learning and deep learning models by adding an explanatory layer to the prediction model [50–52]. This is mainly helpful for investigators and practitioners who operate with various models and need a consistent strategy to explain them.

The SHapley Additive explanations (SHAP) [49] technique is based on a definition of Shapely value presented in cooperative game theory. SHAP notifies the contribution

importance of each feature (factor) in a prediction problem. Besides, SHAP can be deployed to understand how the model generally predicts. Ultimately, it will be utilised to identify and visualise the predictions and each feature's share in the prediction problem. Therefore, SHAP helps the customers notice the details about the prediction and determine how the model is making the prediction. SHAP is a robust instrument to understand how a machine learning model produces predictions. By furnishing insights into the significance of each feature in a prediction problem, SHAP helps to determine potential errors or biases in a model and can also help to enhance the transparency and interpretability of a model.

Besides, SHAP can be utilised to compare the predictions of diverse models and pinpoint which features contribute the most to distinguishing between different categories and classes. One of the preliminary characteristics of the SHAP method is its capability to provide both global and local explanations [54]. Global explanations guide an overall insight into how the model functions, whereas local explanations refer to comprehending how the model made a precise prediction for a specific instance. The SHAP method provides explanations, enabling users to understand the model's behaviour comprehensively.

Furthermore, the SHAP method surpasses addressing complicated dealings between features. In most machine learning methods, the relation between features is unclear and can be challenging to explain. The SHAP method utilises a game-theoretic guideline to compute the assistance of each feature to the model's output, considering all possible combinations of them. This allows end users to gain a deeper understanding of the features' interaction with each other and their contribution to the model's prognoses.

Besides, the SHAP method delivers a suitable infrastructure to interpret a vast range of machine learning models, including linear models, tree-based models, and neural networks. This feature is essential for practitioners and researchers who work with numerous models and require a compatible method to interpret their predictions.

This study serves as an inherent continuation of our preceding work titled "Using AI to predict organisational agility" [12]. In this referring paper, we detailed the prediction process comprising seven machine learning models tailored for predicting organisational agility. In the prediction process, seven classification models were employed, including Support Vector Machine (SVM), K-Nearest Neighbor (KNN), Decision Tree (DT), Random Forest (RF), Gradient Boosting Machines (GBM), Naïve Bayes (NB), and Logistic Regression (LR). The RF achieved the highest test accuracy percentage, confirming the models' effectiveness in predicting organisational agility. The best-performing model was chosen for the model's interpretation.

As a second editorial, this current study delves deeper into the model's explanations using XAI techniques to unveil the underlying factors and dynamics contributing to the prediction decision. Shapley Additive eXplanations (SHAP) was employed in the model's interpretation process. By utilising SHAP, this study unveils the enigmatic realm of predicting organisational agility with a better understanding of how the model generates predictions and allows for more informed decision-making processes on organisational agility/regility prediction issues for scenarios 1 and 2.

This paper aims to improve the agility of small and large organisations through predicting the agility by utilising the random forest machine learning method. Besides, as the main purpose of the paper, the SHAP method has been adopted to not only explain and highlight the most important features in the organisational agility prediction problem but to identify the least important features which can be replaced in the future with potentially overlooked features in predicting organisational agility. Furthermore, the study investigates whether the features yield positive, negative, or both effects on the prediction problem. XAI techniques

improve the reliability of AI systems by ensuring that they generate accurate and complete explanations for their decisions.

The structure of the paper is organised as follows. Section 2 demonstrates the material and methods, encompassing the data used in this study, an overview of the prediction process, and interpretation techniques implemented in this study. Section 3 presents and discusses the results and findings obtained from the XAI techniques in the interpretation process. Section 4 concludes the study by summarising its findings, limitations, and suggestions for future directions.

## 2. Materials and methods

### 2.1 Data collection

The data that was collected for this research is sourced from 44 industry practitioners from differing industry sectors in Australia (with 42% in private sector organisations, 17% public sector, 22% in services organisations, 11% in Universities, and 8% not for profit and others) using an online questionnaire survey with a semi-structured questionnaire (The Human Research Ethics Committee of Torrens University Australia approved this study "Project title: Supporting project success in Australia with the implementation of project portfolio management." (HREC Reference number: EAN 2015 09 (6)). The Data Collection was conducted between 30 Sep 2015 to 30 Feb 2016.

There has been a diverse range of organisation types and sizes in the sample size covered below.

The results are based on the primary roles of the respondents (refer to Fig 1) were Project Manager (19%), Team Member (16%), Manager (13%), Executive General Manager (13%), and business trainers or lecturer roles within their organisation (19%).

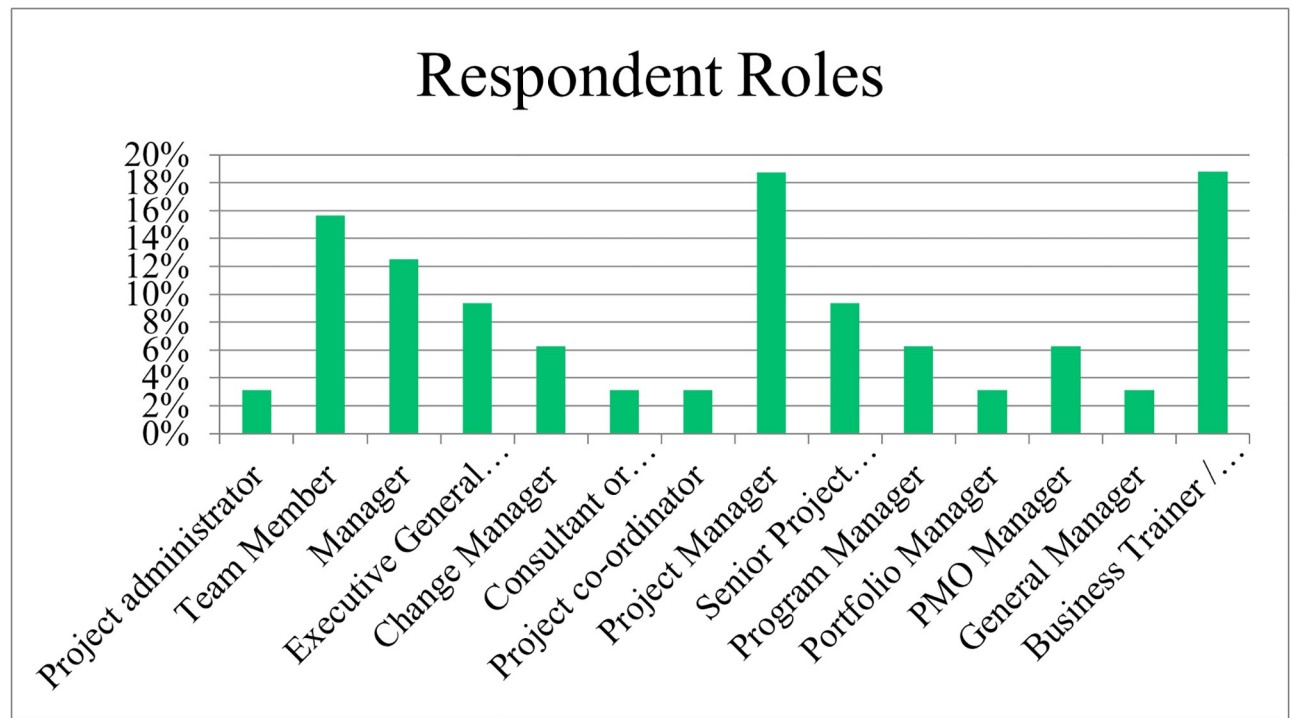

**Fig 1. Respondent roles.**

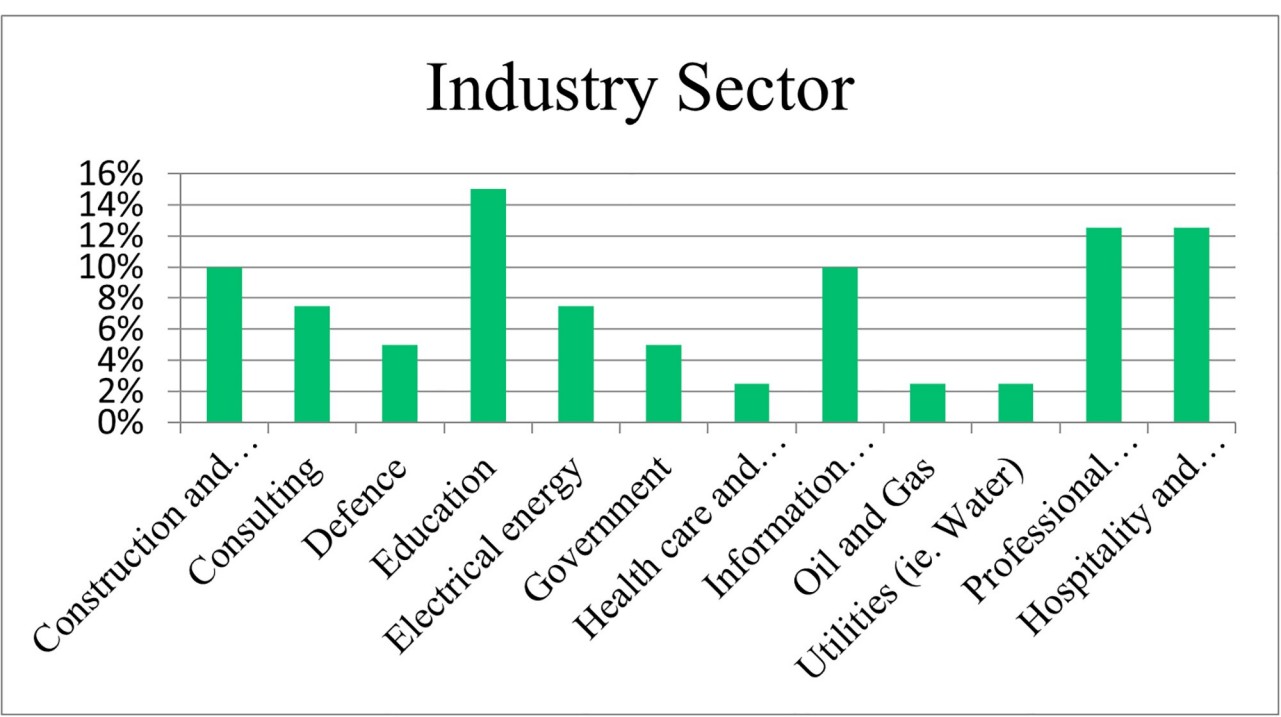

**Fig 2. Respondent industry sector.**

The predominant industry sector was education (15%), followed by Professional Services, Hospitality and Tourism (13%) (refer to Fig 2).

In Fig 3, a significant portion (33%) of respondents were from organisations with a size in the 300 to 2000 range, with 17% less than 19 and from 100 to 299 employees (refer Fig 3).

## 2.2 Prediction process of Random Forest algorithm

Adequate data is required to understand the connections between input and output attributes. Data analysts confront limited data, even by knowing the significance of data. Therefore, Synthetic Data Vault (SDV) in Python package [55] was utilised for the current data to breed additional synthetic data in the investigation. SDV is a well-known synthetic data generator that validates the infrastructure using Generative Adversarial Networks (GANs) or multidimensional accumulative distribution functions. The SDV system requires four sequential steps to complete its task of generating additional synthetic data [55], including organising, specifying structure, learning model, and synthesising data.

Phase one, Organise: Before providing the dataset to the SDV, the user should arrange the database's data into distinct files.

Phase two, Specify Structure: The user should pinpoint fundamental information about the configuration of each table and supply it as metadata for the database. This procedure resembles a framework in an SQL database. Columns with ID knowledge are unique because they include connected information between considerable tables. The user should pinpoint that table when the ID column of a table references an ID column of a further table.

Phase three, Learn Model: The user gathers the SDV's script to comprehend the generative prototype. The SDV repeats via tables consecutively, utilising a modelling algorithm devised to account for connections between the tables. For each table, the SDV uncovers a configuration

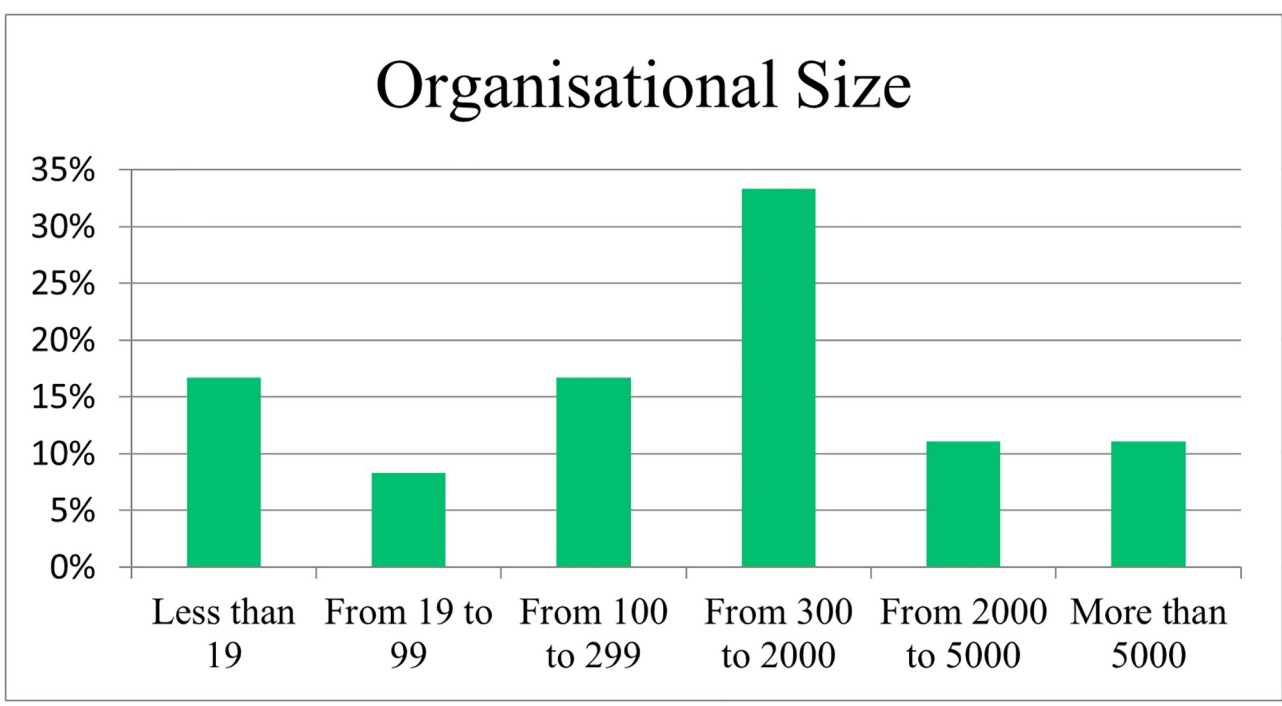

**Fig 3. Respondent organisational size.**

of reliance (dependence). If different tables reference the existing one, reliance exists, and the SDV calculates the gathered statistics for different tables. The collected statistics are then counted into the initial table, constructing a developed table. This generated table is then formed. It catches the generating knowledge for the initial table columns and all the reliance between tables. The SDV utilises some uncomplicated optimisation to enhance efficiency. It stores all the developed tables and prototype knowledge in exterior files to not execute the same unnecessary calculations for the duplicated database.

The user encounters a straightforward API with three principal parts after representing the SDV to a database.

1. database.get_table: This yields a model for a distinct table in the database. When the table has been discovered, the user utilises it to fulfill the other two parts.

2. table.synth_row: The synth_row process both synthesises rows and skips data.

3. table.synth_children: The synth_children process synthesises entire tables that reference the existing table. The user synthesises the whole database by employing the function repetitively on newly-synthesised tables. Both synth_children and synth_row conclusions match the initial data precisely. The SDV takes phases to omit comprehensive data, round weights, and generate spontaneous text for the textual columns.

### 2.3 Experiment procedure

The data that has been collected for this study came from 44 industry participants from various sectors of industry in Australia (42% from the private sector, 17% from the public sector, 22% from the service organisation, 11% from the Universities, and 8% not for profit and others)

that was an online survey applying a semi-structured questionnaire using closed questions provided clear (yes/no) and specific answers. On the other hand, additional information or comments from the participants that are not included in the closed question can be added to others. Most domains in the data consist of definite variables. The questionnaire inspection participants were asked for their role within the organisations, country, the number of workers within the association, the organisational type, respondent industry sector, whether the organisation had a PMO, agile characteristics within organisation, organisational size, aspects of future change, agile and practices of the organisational agility, level of maturity, and levels of practices within the organisation, changes that happened within the last two years, exterior elements that changed, perceived barriers and benefits to enhance the organisational agility and facets of maturity to sustain organisational agility.

Organisational Agility is the capability to promptly adjust to changes and perform necessary actions to keep up the growth rate. It requires a variety of attributes and behaviours, for instance, decentralised and quick decision-making; active, ambitious, and agile group members; self-aware, ethical, and customer-oriented skill development; motivational leadership and ongoing learning from experience.

A precise, high-quality, complete, and enormous dataset is necessary to adequately instruct a machine learning algorithm, as without it, even the most practical ML model may abort. Therefore, it is essential to have a thoroughgoing dataset to supply proper and accurate predictions.

Machine learning links the input data to the output data. To provide enough data, Synthetic Data Vault in Python package was used to find out the relation between inputs and outputs. The data was managed based on the scenarios (scenario 1, scenario 2). The scenarios include three classes of outcomes about the degree of organisational agility (classes 1, 2, or 3).

The dataset used in this research is based on 21 factors associated with organisational agility and project portfolio management, which have been ordered into three distinctive data groups (i.e., Group 1, Group 2, and Group 3) by using Canonical analysis after a comprehensive review of synthetic data. In the end, 94 features for scenario 1 and 142 features for scenario 2 have been ranked by the SHAP method.

RF algorithm had better performance in comparison with other algorithms such as SVM, GBM, DT, LR, and NB to predict the organisational agility [12]. Meanwhile, with the test accuracy of 97.67%, the RF algorithm is ranked higher than other algorithms for agility prediction.

On the other hand, the system acts like a black-box, and the features' importance shared is unknown. The SHAP method has been deployed against this problem to overcome this obstacle.

Learning algorithms that scale with the volume of information are needed to exploit the size of trendy datasets while holding adequate statistical effectiveness. The random forest (RF) method, developed by L. Breiman [56], is a portion of the index of the most prosperous methods available nowadays to manage datasets. RF is a supervised learning technique affected by the premature attempts of [28, 29, 57], functions based on the uncomplicated but effective "divide and conquer" principle, which estates sample fractions of data, develops a randomised tree anticipator on a tiny piece, and then aggregates these anticipators together.

Forests can be applied to a broad spectrum of prediction issues and have rare parameters to adjust, which is the reason for the reputation of forests. Furthermore, the method is acknowledged for its delicacy and capability of dealing with short sampling sizes [58]. Simultaneously, there is the prospect of facing gigantic actual systems.

This study uses the random forest method to predict organisational agility for two scenarios. Fig 4 illustrates a random forest algorithm flowchart to solve a prediction problem.

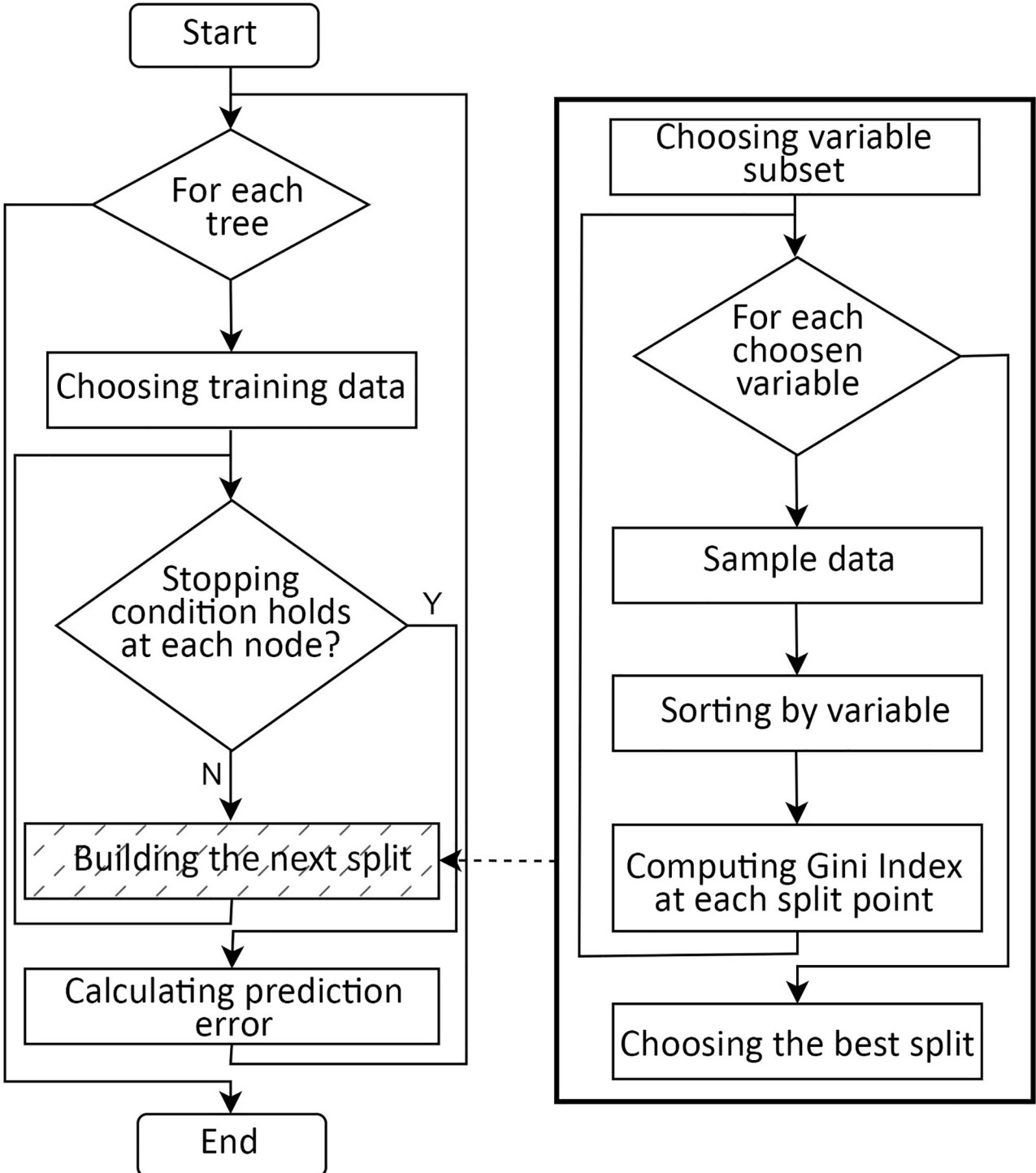

**Fig 4. The flowchart of the Random Forest algorithm.**

## 2.4 Model's interpretation

The SHapley Additive explanations (SHAP) [49] technique was established based on a characterisation of Shapely value introduced in cooperative game theory. As mentioned earlier, the SHAP method is a formidable tool for comprehending the feasibility of prediction in a machine learning model. By furnishing insights into the significance of each attribute in a prediction problem, the SHAP method can help to pinpoint biases or errors in the model.

It can also contribute to enhancing the transparency and explainability of the model. Besides, the SHAP method can be deployed to compare the different models' predictions. This is conducted by merging each potential subset of parties and reassessing the outcome. Lloyd S. Shapley suggested the ensuing explanation formula for Shapley values.

$$\varphi_j(f, x) = \sum_{Z' \subseteq x'} \frac{|Z'|!(M - |Z'| - 1)!}{M!} \left[ f_x(Z') - f_x\left( {z'}/{j} \right) \right] \tag{1}$$

The purpose of developing this for a machine learning problem is to monitor how the various features affect the model's output. From Eq (1), by employing the model f and attribute vector x, the Shapley value of attribute j is calculated. As the majority of models are complicated, simplifying the attributes into X' is needed.

The foremost part of Eq (1) is a weight that illustrates how omitting or adding diverse features affects the model based on all features. $[f_x (z-)-f_x (Z' /j)]$ declared the contribution of features subset, $i$ in $z'$, where $f_x (z') = E[f(z)|Zs]$. A set S is offered by non-zero indices in $z'$ [56]. As Shapley values are computed on all subsets of contributors, the numbers of computations increased significantly.

Fig 5. shows a brief schematic of the data generation process of the study's eXplainable AI algorithm, SHAP.

# 3. Result and discussion

The COVID-19 pandemic has highlighted the importance of organisational agility and resilience. Organisations that are able to adapt quickly to change, anticipate and prepare for disruptions, and respond and recover effectively are more likely to succeed in the current volatile business environment. By understanding organisational agility, organisations will improve their organisational regility. AI can play a significant role in improving organisational agility and resilience. This section presents SHAP results based on RF algorithm results for scenarios 1 and 2.

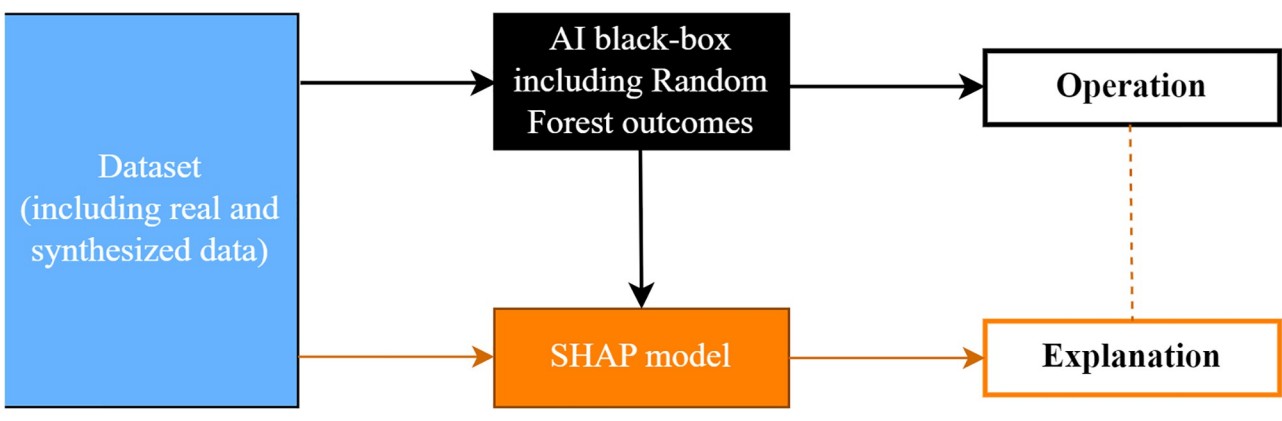

**Fig 5. AI explanation chart.**

Fig 6 illustrates the SHAP summary plots for scenario 1 organisational agility prediction with the RF algorithm. In Fig 6(a), the SHAP summary plot illustrates the variables' contribution order on organisational agility prediction for scenario 1 with the RF algorithm. The features' importance has been presented in descending order with the SHAP values and features on the x-axis and Y-axis, respectively. As it is presented, legislation, globalisation, low level of process maturity, technology and digital disruption 1, competitors, technology and digital disruption, market size 1, market share, organisational type, and organisational sustainability are the most important ten features that affect the organisational agility prediction. Conversely, features such as lack of buy-in, market size, legacy systems (IT, manufacturing, etc.), and changes in the external environment are the less effective ones on organisational agility prediction.

In Fig 6(b), the data samples are introduced as coloured dots from the blue colour to the red colour in the original values of the sequence of the features. To make it visually clear, the most important and least important attributes have been shown in Fig 6(c). The red distribution represents higher feature values within the observation, while the blue represents the prevalence of lower feature values. Each dot and its colour represent an industry practitioner and its feature value, respectively. Regardless of the colours, the dots that have been located in negative values of the x-axis have a negative impact on our prediction problem, and the dots that have been located in positive values of the x-axis have a positive impact on prediction. Therefore, the amount of negativity and positivity of the SHAP value, along with the colour, has set the features' importance for each company practitioner and all companies together.

This information can be used by researchers and practitioners to gain a deeper understanding of the factors influencing organisational agility and to develop strategies for enhancing it for scenario 1. By understanding the feature importance of organisational agility, organisations and companies will find the attributes that affect organisational regility, too, as these two organisational factors are correlated with each other closely.

The identification of influential features can assist organisations in understanding which aspects they should prioritise and focus on to enhance their agility and regility. For instance, organisations can potentially improve their agility by addressing issues related to legislation, globalisation, Low level of process maturity, and technology and digital disruption. Conversely, the less important features, such as lack of buy-in, market size, legacy system (IT, manufacturing, etc.), and changes in the external environment, may require less attention in improving organisational agility prediction.

Fig 7 visually depicts the results derived from the SHAP analysis for scenario 1. The primary objective of this analysis is to clarify the relative importance of various input features concerning the prediction outcomes within three distinct classes. Within the plot, the mean absolute SHAP values are assigned to the twenty most influential features on prediction across these classes. To comprehend the significance of these characteristics, a colour-coded scheme is implemented: the presence of green signifies features influencing class 0 predictions, blue denotes those impacting class 1 predictions, and pink represents features pertinent to class 2 predictions within scenario 1. The essence of this analysis is conveyed through the varying lengths of the bars corresponding to each feature. Longer bars signify a heightened importance in predicting their respective classes, while shorter bars signify a relatively lower impact on prediction outcomes. Moreover, higher SHAP values underscore the significance of these features in the analysis.

SHAP has an excellent ability to explain the dynamics of values that correspond to changes in individual feature values in organisational agility. Fig 8 provides a visually informative dependence plot generated using the SHAP technique, focusing on single-feature analysis for scenario 1. In this particular context, we focus on examining complex interactions involving

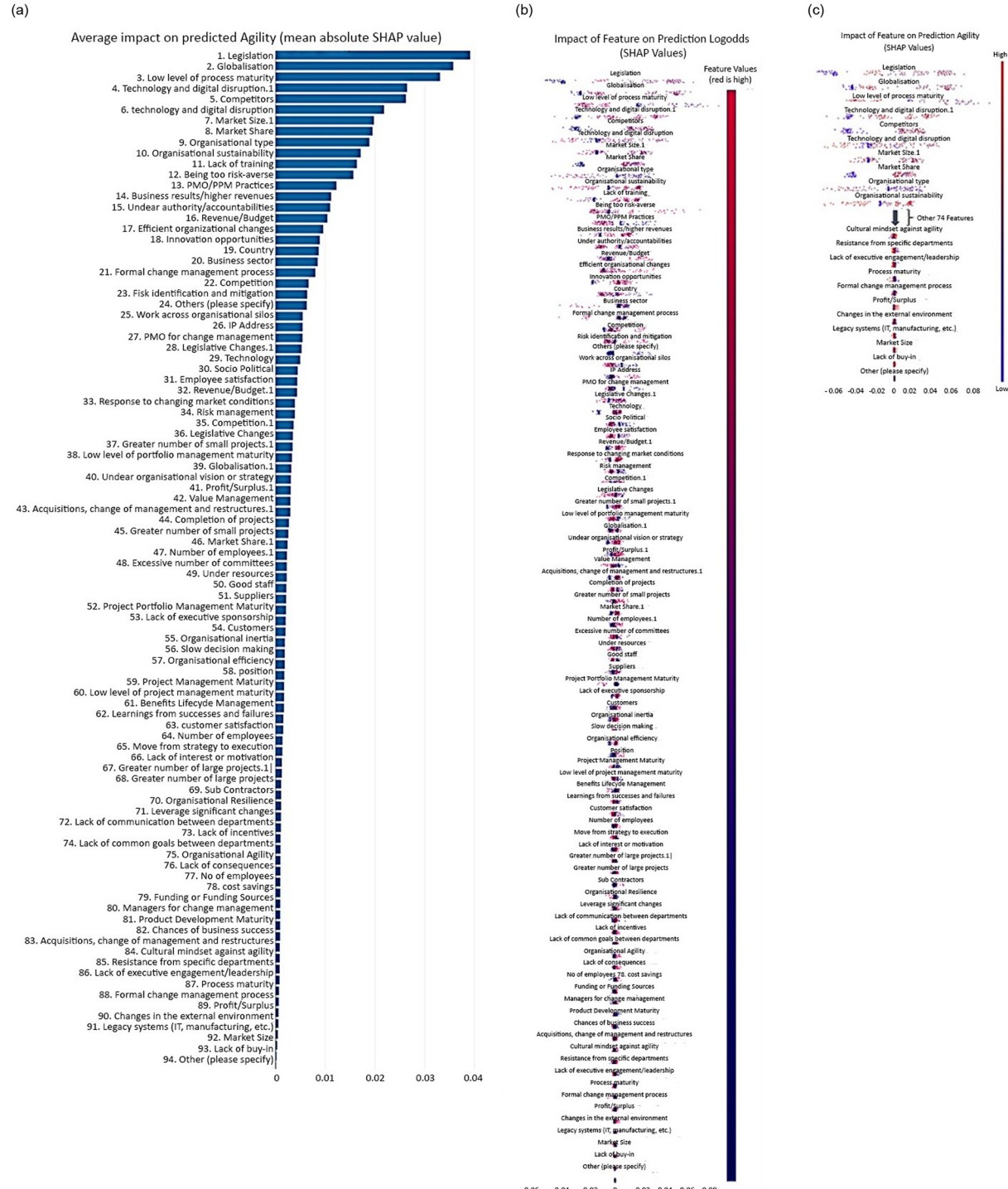

**Fig 6.** SHAP summary plots for Scenario 1 (a) demonstrate the order of importance of the ten features according to the mean (SHAP value); the higher the SHAP value of a trait is given; (b) illustrates the interpretation of the RF model, ranking the importance of the 94 features in descending order. (c) Shows the magnifiedFig 6(b) considering the ten most important features and 10 least important features.

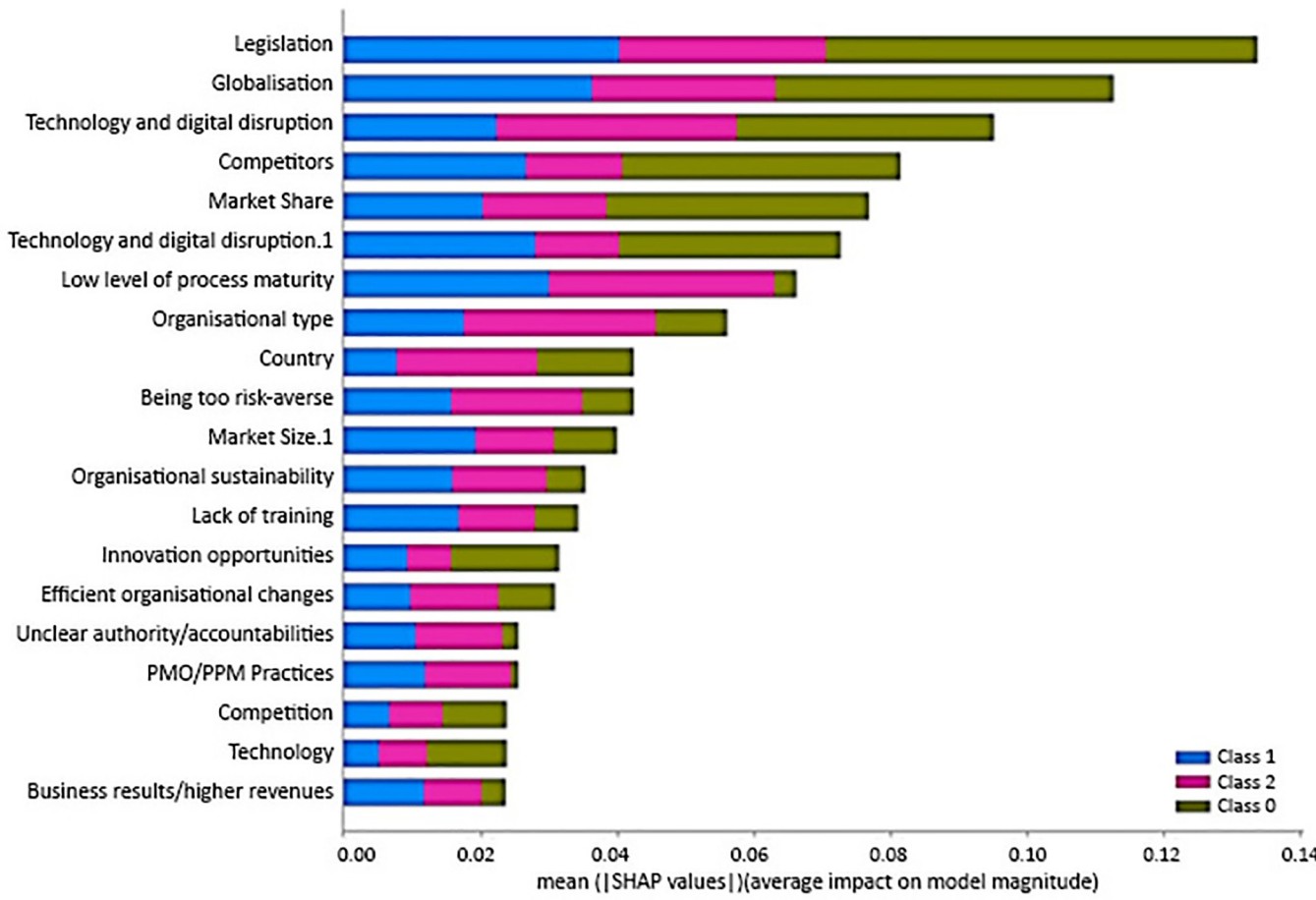

**Fig 7. Summary plot for SHAP analysis shows the mean absolute SHAP value of the twenty most influential features on predicting organisational agility for the three classes for scenario 1.**

the "Legislation" attribute, which arguably emerges as the most influential factor exerting its influence on the prediction outcome generated by RF. Existing analytical efforts include a wide range of features in data, including those with significant influence and those with minimal impact on the overall dataset. Therefore, our investigation is aimed to investigate the correlation indicated by the attribute "Legislation" in relation to some of the selected most influential features, including globalisation, low level of process maturity, technology and digital disruption 1, and competitors. In order to visually represent complex relationships, the graphical approach uses two axes. The SHAP value for a specific data point relative to the legislation feature is plotted along the left vertical axis. In contrast, the corresponding globalisation values are plotted along the horizontal axis.

The inherent functionality of the SHAP technique is exploited to identify the attribute that is most prominently associated with the legislation feature. These distinguished relations, which in this instance is globalisation, which illustrated along the right vertical axis as depicted in Fig 8a. The visualisation is further enriched using a colour spectrum, where the selected interaction attribute -globalisation in this scenario- is dedicated to the colour assigned to each data point. Fig 8a–8d are related to the influence of the dependence plot for legislation with globalisation, low level of process maturity, technology and digital disruption 1, and

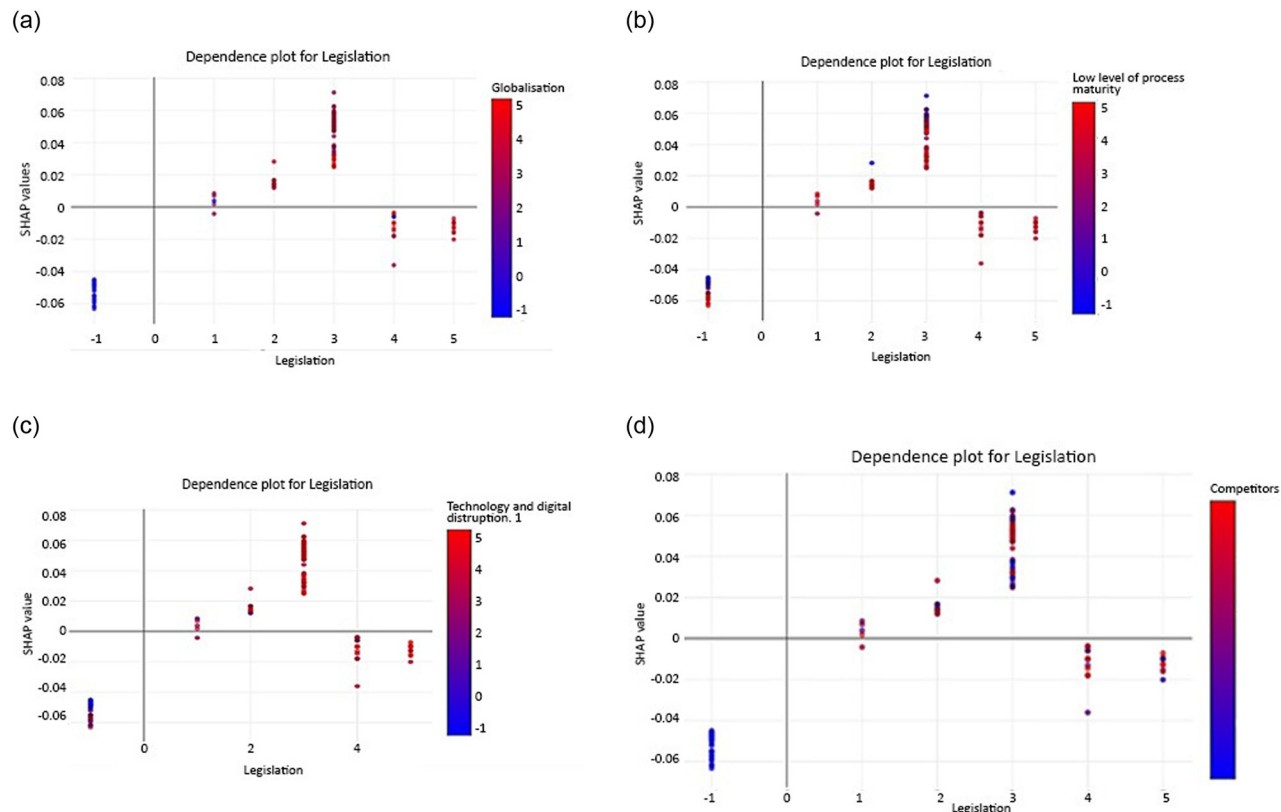

**Fig 8.** SHAP dependency plot for scenario 1, (a) globalisation and legislation, (b) globalisation and low level of process maturity, (c) globalisation and technology and digital disruption 1, and (d) globalisation and competitors.

competitors on the prediction of organisational agility. The legislation presents both negative and positive effects on organisational agility. The impact of the department ranges from -1 to 5, fluctuating from -0.06 to +0.08.

On the other hand, low and high legislation values have a negative impact on organisational agility. The fact that the legislation feature has both positive and negative effects on prediction outcomes indicates that different legislation has different impacts on organisational agility. Some legislation values, ranging from 1 to 3, have a positive impact on prediction. The relationship between legislation and globalisation on organisational agility exhibits a non-linear relationship. This suggests that the influence of legislation on organisational agility may not be consistent across all feature values and may have different effects depending on other factors. As can be seen, the two elements interact, showing different colours than red or blue. In this case, the features are shown to have dependent and overlapping effects on the model output. Overlapping dots indicate that there may be cases where different departments have the same or similar values for this feature, but their impact on prediction varies. This may be due to interactions with other features that affect the outcome.

Fig 9 presents the feature importance sequence in the organisational agility prediction problem for scenario 2 with the RF algorithm by the SHAP summary plot. Globalisation, legislation, technology and digital disruption 1, self-aware and honest, open communication, market share, competition 1, Iterative/incremental PHP, and organisational type are the most effective ten features in organisational agility prediction problems. The lack of buy-in, lack of

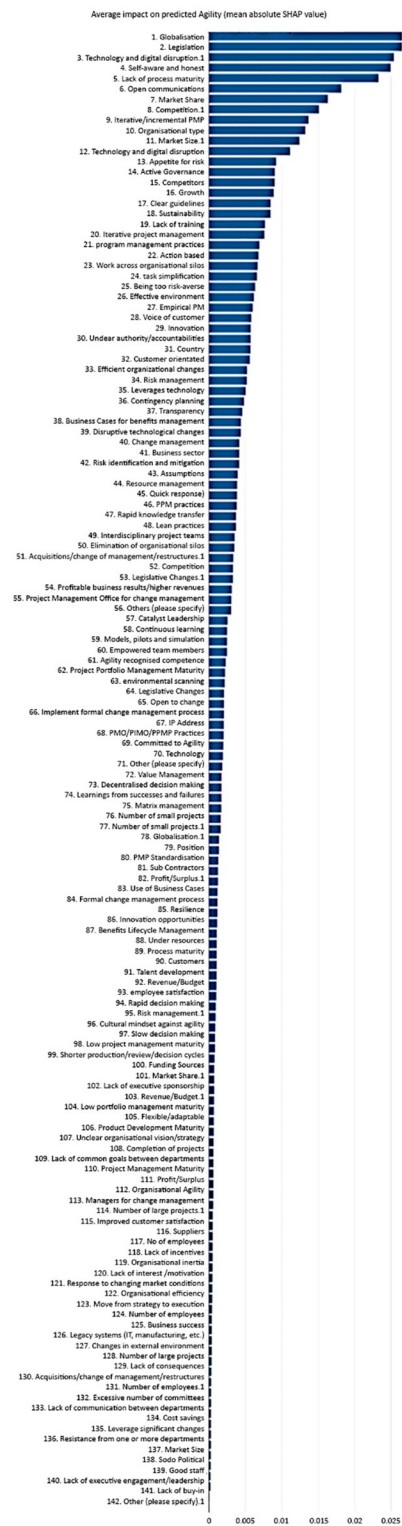

**Fig 9. The SHAP summary plot for scenario 2 demonstrates the order of importance of the ten features according to the mean (SHAP value); the higher the SHAP value of a trait is given.**

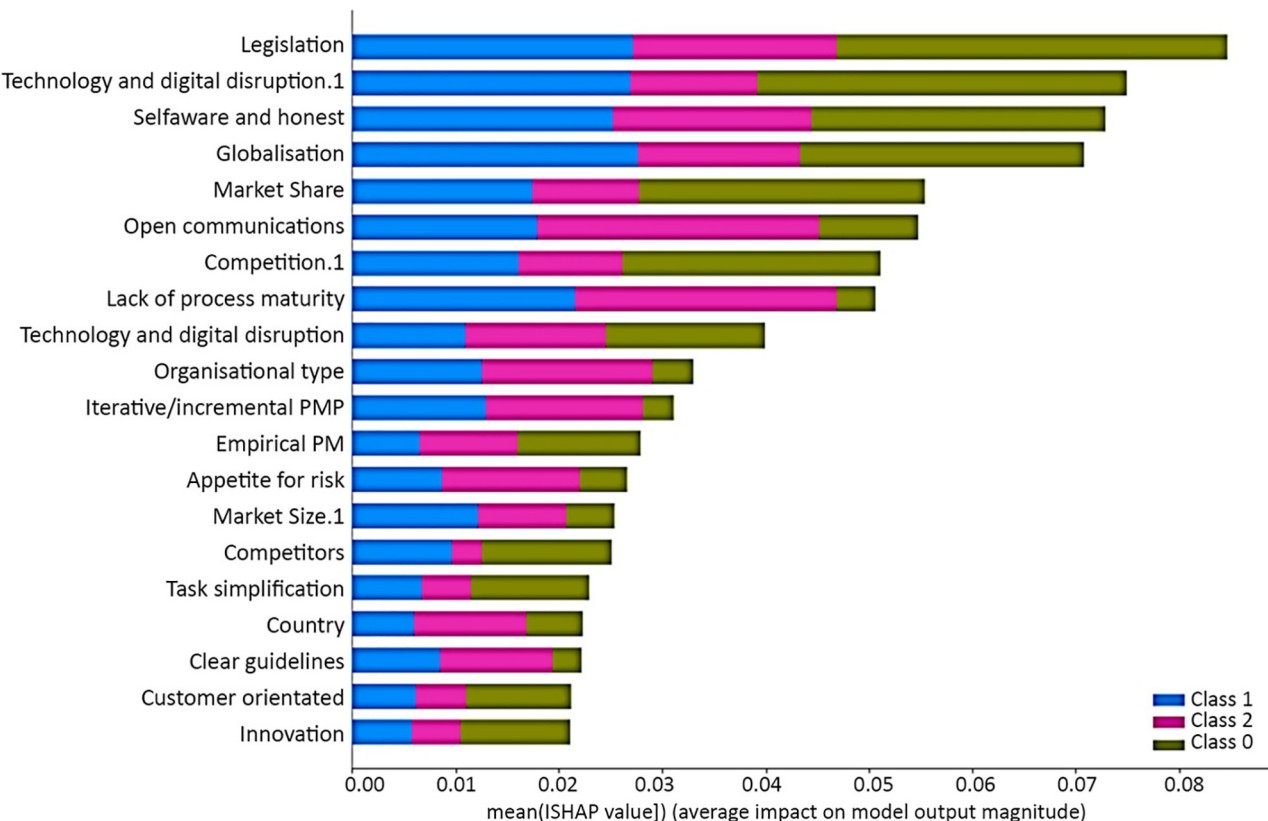

**Fig 10. Summary plot for SHAP analysis shows the mean absolute SHAP value of the twenty most influential features on predicting organisational agility for the three classes for scenario 2.**

executive engagement/leadership, good staff, socio-political, and market size are the least influential features on the organisational agility prediction problem for scenario 2. By prioritising and focusing on the most influential factors, they can better position themselves to achieve higher levels of organisational agility and regility. Fig 10 visually depicts the results derived from the SHAP analysis to clarify the relative importance of different input features in prediction outcomes in three classes for scenario 2.

Fig 11 provides a visually informative dependence plot generated using the SHAP technique, focusing on single-feature analysis for scenario 2. As mentioned earlier, globalisation, legislation, technology, and digital disruption.1, self-aware and honest, and lack of process maturity are the most efficient contributors to organisational agility prediction for scenario 2. Therefore, a SHAP dependency plot is utilised to evaluate the SHAP value dependency on each feature for scenario 2 as well. Fig 11a–11d are related to the influence of the dependence plot for globalisation with legislation, technology, and digital disruption 1, self-aware and honest, and open communication on the prediction of organisational agility. The legislation presents both negative and positive effects on organisational agility. The impact of globalisation ranges from -0.05 to 0.04.

While this study provides insights, it has some limitations that provide opportunities for further exploration. First, it is essential to note that our study was conducted with a relatively small data sample. Therefore, the applicability of our findings to a broader range of institutions may be somewhat limited. Second, our research was conducted in a relatively short time frame

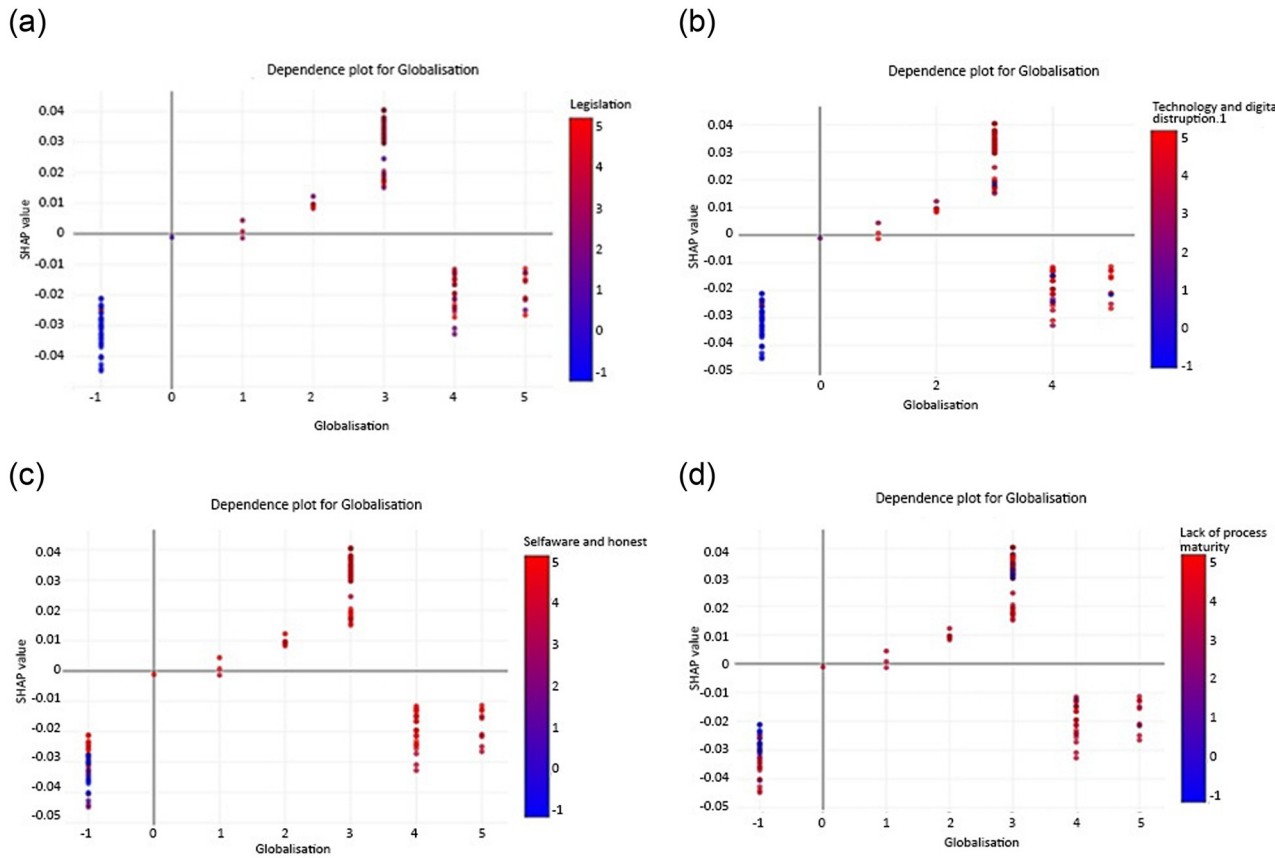

**Fig 11.** SHAP dependency plot for scenario 2, (a) globalisation and legislation, (b) globalisation and technology and digital disruption 1, (c) globalisation and self-aware and honest, and (d) globalisation and lack of process maturity.

before the pandemic. Given the potential impact of such a significant global event on organisational dynamics, it is essential to recognise that responses and outcomes may differ in a post-pandemic context. Additionally, our study primarily focused on the use of AI as a means of improving organisational agility and resilience. However, it should be recognised that various other contributing factors, such as organisational culture, and external factors, such as environmental or political influences, play a role in shaping these aspects.

Despite these limitations, our research opens up promising avenues for future investigations in the field of AI and organisational agility and resilience. First, broadening the scope of the research to cover different institutions would help ensure the generalisability of our study findings. This broader approach will help determine whether similar trends emerge in different organisational contexts. Second, conducting longitudinal studies over a long period of time will provide insight into how the implementation of AI technology affects the agility and resilience of organisations over time, allowing us to track its evolution and impact. Finally, future researchers can explore the interplay between AI and other influential factors, such as organisational culture, environmental conditions, and political dynamics, to better understand what contributes to agility and organisational resilience in a complex and dynamic world. These multidisciplinary investigations will enrich our knowledge and inform more holistic approaches to improving organisational effectiveness.

## 4. Conclusion

As a second editorial, this study is a sequel to our preceding work [12]. In the referring paper, the prediction process employed seven classification models, including SVM, KNN, DT, RF, GBM, NB, and LR. The RF achieved the highest test accuracy percentage, confirming the models' effectiveness in predicting organisational agility and being chosen for the model's interpretation process using XAI techniques. To build trust and transparency in AI systems, An XAI method has been adopted to provide precise and comprehensive explanations of how decisions are made and how different features influence the organisational agility. Therefore, in this study, the SHAP method has been deployed to explain the black-box and clarify each feature's importance share on the problem of organisational agility prediction for the two scenarios.

The top five most effective features which affect the organisational agility prediction are legislation, globalisation, low level of process maturity, technology and digital disruption, and competitors for scenario 1 and also globalisation, legislation, technology and digital disruption.1, self-awareness, honesty, and lack of process maturity for scenario 2, respectively. Conversely, five less effective features in predicting organisational agility are lack of buy-in, market size, legacy systems, changes in the external environment, and profit/surplus for scenario 1 and lack of buy-in, lack of executive engagement/leadership, good staff, socio-political, and market size for scenario2, respectively. Our aspiration is for our findings to assume a central role in facilitating organisations' endeavour to augment their agility, consequently empowering them to secure a competitive advantage amidst the fluid and perpetually evolving business environment.

Researchers can continue to investigate the potential of XAI in improving organisational regility by considering probably overlooked features using different explainability methods such as LIME, GRAD-CAM, and DeepLIFT in the future.

## Author Contributions

**Conceptualization:** Niusha Shafiabady, Nick Hadjinicolaou.

**Data curation:** Niusha Shafiabady, Nick Hadjinicolaou.

**Formal analysis:** Niusha Shafiabady.

**Investigation:** Niusha Shafiabady.

**Methodology:** Niusha Shafiabady, James Vakilian.

**Project administration:** Niusha Shafiabady.

**Resources:** Nick Hadjinicolaou, Robert M. X. Wu.

**Software:** Niusha Shafiabady.

**Supervision:** Niusha Shafiabady.

**Validation:** Niusha Shafiabady.

**Visualization:** Niusha Shafiabady, Nick Hadjinicolaou.

**Writing – original draft:** Nick Hadjinicolaou, Nadeesha Hettikankanamage, Ehsan MohammadiSavadkoohi.

**Writing – review & editing:** Niusha Shafiabady, Nadeesha Hettikankanamage, Robert M. X. Wu, James Vakilian.

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
