## [Decision Letter · Decision Letter 0]

17 Jan 2024

PONE-D-23-32213eXplainable Artificial Intelligence (XAI) for improving organisational regilityPLOS ONE

Dear Dr. Shafiabady,

Thank you for submitting your manuscript to PLOS ONE. After careful consideration, we feel that it has merit but does not fully meet PLOS ONE’s publication criteria as it currently stands. Therefore, we invite you to submit a revised version of the manuscript that addresses the points raised during the review process.

**ACADEMIC EDITOR: based on the  reviewers comments, the paper needs more improvements by incorporating the recommendation appended below. Hence my  decision is Major Revision **==============================

We look forward to receiving your revised manuscript.

Kind regards,

Shahid Akbar, PhD

Academic Editor

PLOS ONE

Reviewers' comments:

Reviewer's Responses to Questions

**Comments to the Author**

1. Is the manuscript technically sound, and do the data support the conclusions?

Reviewer #1: Yes

Reviewer #2: Partly

2. Has the statistical analysis been performed appropriately and rigorously? 

Reviewer #1: Yes

Reviewer #2: Yes

3. Have the authors made all data underlying the findings in their manuscript fully available?

Reviewer #1: Yes

Reviewer #2: Yes

4. Is the manuscript presented in an intelligible fashion and written in standard English?

Reviewer #1: Yes

Reviewer #2: Yes

5. Review Comments to the Author

Reviewer #1: 1. At the end of introduction, the main contribution of the proposed study should be provided in points.

2. The quality of all the figures are very poor, authors are advised revising all the figures in 600 dpi for clear visualization.

3. To show the effectiveness of the proposed study, the proposed results should be compared with the existing studies.

4. The authors are advised to provide the hyperparameters used for training the applied machine learning models by citing the recent predictors such as DP-BINDER, pAtbP-EnC, AIPs-SnTCN, and Target-ensC_NP .

5. In developing machine learning based models, the validation of the model is an important phase, the authors should make it clear that they deals with overfitting issues of the proposed study

Reviewer #2: To improve the quality of the manuscript. The following recommendations need to be incorporated.

The authors are suggested to provide up-to-date related work with critical analysis.

The proposed study was performed on the old data of the early 2015-16. It would be more convenient if the authors applied the methodology to the latest data.

The conclusion seems very lengthy and not attractive. Authors are advised to revise the conclusion section concisely and briefly.

What should be the real-life applications of the proposed study? Moreover, the future directions should be provided.

The authors are advised to proofread the whole manuscript for typos and grammatical mistakes

6. PLOS authors have the option to publish the peer review history of their article (what does this mean?). If published, this will include your full peer review and any attached files.

Reviewer #1: No

Reviewer #2: No

---

## [Author Response · Author response to Decision Letter 0]

27 Feb 2024

Table 1: Reviewers’ Responses

Item no Response

Reviewer1 We would like to thank Reviewer 1 for their excellent comments. We do appreciate your comments, they were very helpful. We have modified and restructured the mentioned sections accordingly and addressed your comments. 

Reviewer2 We would like to thank Reviewer 2 for their great comments. We have addressed the comments in detail within the updated version of the manuscript. 

Table 2: Detailed Reviewers’ Responses

Item No 

Reviewer 1 Reviewer #1: eXplainable Artificial Intelligence (XAI) for improving organisational regility

1) At the end of introduction, the main contribution of the proposed study should be provided in points.

Response: Thanks for the comment. A paragraph is added to the introduction to emphasise the importance of eXplainability to this paper.

2) The quality of all the figures are very poor, authors are advised revising all the figures in 600 dpi for clear visualization.

Response: The authors appreciate the time and effort of the reviewer and this constructive comment. The images have been modified and amended. 

3) To show the effectiveness of the proposed study, the proposed results should be compared with the existing studies. 

Response: Thanks for the comment. As outlined in the introduction, this work builds upon our previous research on prediction of organisational agility. In the previous work, we focused on the prediction concepts and algorithms. 

In this work, however, we aimed the eXplainability. As you astutely mentioned in comment 1, we pinpointed the aims of the work in a nutshell to clarify it, in the introduction section. Based on the manuscripts’ results, the concept of explainability is relative and relies on the specific features we choose for our prediction problem. Therefore, It cannot be compared to other pieces of literature in this field unless all features are identical. Besides, using explainibility in the prediction of organisational ability is the main innovations of our work. 

Therefore, we will add the accuracy of the random forest method as the method with the highest accuracy based on our previous study, as it is missing in the manuscript. Besides, three paragraphs are added to show the research gap, in the introduction.

4) The authors are advised to provide the hyperparameters used for training the applied machine learning models by citing the recent predictors such as DP-BINDER, pAtbP-EnC, AIPs-SnTCN, and Target-ensC_NP.

Response: The authors thank the reviewer for the deep look into the manuscript. Above mentioned parameters are related to the prediction algorithm used in previous study, and we have addressed things like hyperparameters in previous work (reference 12 of the manuscript).

5) In developing machine learning based models, the validation of the model is an important phase, the authors should make it clear that they deals with overfitting issues of the proposed study.

Response: Thank you for the comment. The focus of this work is on the explainability methods and the results demonstrated in the study are basically based on the validation sets.

Reviewer 2 Reviewer #2: eXplainable Artificial Intelligence (XAI) for improving organisational regility.

1) The authors are suggested to provide up-to-date related work with critical analysis. 

Response: Thanks for your comment. Three paragraphs are added to the introduction to consider some other related references and also show the gap. We also added one paragraph to the introduction to illustrate the main purpose of the paper clearly.

 2) The proposed study was performed on the old data of the early 2015-16. It would be more convenient if the authors applied the methodology to the latest data.

Response: Thanks for the comment. The data used in the manuscript is not time-dependent and it has been collected from Australian business owners and it is using emerging technology. Besides, it is the continuation of the previous work. 

3) The conclusion seems very lengthy and not attractive. Authors are advised to revise the conclusion section concisely and briefly. 

Response: The authors want to thank the reviewer because this comment helped the manuscript’s consistency. The conclusion has been rewritten and modified.

4) What should be the real-life applications of the proposed study? Moreover, the future directions should be provided. 

Response: Thanks for the comment. About the real-world application of the paper results, as stated in the conclusion, “. Our aspiration is for our findings to assume a central role in facilitating organisations’ endeavour to augment their agility, consequently empowering them to secure a competitive advantage amidst the fluid and perpetually evolving business environment”. Besides, as you wisely mentioned this issue, we added some sentences in the manuscript to clarify the importance of the manuscript’s finding in real-world application mentioning that the findings of this paper help the organisations understand the most and least important factors which affect the organisational regility and try to focus on the most important ones to improve it. Organisations can also replace the least important features with any upcoming feature during the time and optimise their resiliency and agility.

About the future direction of the work, a paragraph is added to conclusion. 

5) The authors are advised to proofread the whole manuscript for typos and grammatical mistakes

Response: Thank you for the helpful comment. The whole manuscript has been grammar-checked and the issues have been addressed.

---

## [Decision Letter · Decision Letter 1]

17 Mar 2024

eXplainable Artificial Intelligence (XAI) for improving organisational regility

PONE-D-23-32213R1

Dear Dr. Shafiabady,

We’re pleased to inform you that your manuscript has been judged scientifically suitable for publication and will be formally accepted for publication once it meets all outstanding technical requirements.

Kind regards,

Shahid Akbar, PhD

Academic Editor

PLOS ONE

Additional Editor Comments (optional):

Reviewers' comments:

Reviewer's Responses to Questions

**Comments to the Author**

1. If the authors have adequately addressed your comments raised in a previous round of review and you feel that this manuscript is now acceptable for publication, you may indicate that here to bypass the “Comments to the Author” section, enter your conflict of interest statement in the “Confidential to Editor” section, and submit your "Accept" recommendation.

Reviewer #1: All comments have been addressed

Reviewer #2: All comments have been addressed

2. Is the manuscript technically sound, and do the data support the conclusions?

Reviewer #1: Yes

Reviewer #2: Yes

3. Has the statistical analysis been performed appropriately and rigorously? 

Reviewer #1: Yes

Reviewer #2: Yes

4. Have the authors made all data underlying the findings in their manuscript fully available?

Reviewer #1: Yes

Reviewer #2: Yes

5. Is the manuscript presented in an intelligible fashion and written in standard English?

Reviewer #1: Yes

Reviewer #2: Yes

6. Review Comments to the Author

Reviewer #1: yes, my previous concerns are successfully incorporated in the revised version. no further comments.

Reviewer #2: All changes are incorporated. No further addition is required. It is in good form and presented in a better way

7. PLOS authors have the option to publish the peer review history of their article (what does this mean?). If published, this will include your full peer review and any attached files.

Reviewer #1: No

Reviewer #2: No

---

## [Editor Report · Acceptance letter]

25 Mar 2024

PONE-D-23-32213R1 

PLOS ONE

Dear Dr. Shafiabady, 

I'm pleased to inform you that your manuscript has been deemed suitable for publication in PLOS ONE. Congratulations! Your manuscript is now being handed over to our production team.

Kind regards, 

on behalf of

Dr. Shahid Akbar 

Academic Editor

PLOS ONE